# Non-Smoker’s Exposure to Second-Hand Smoke in South Africa during 2017

**DOI:** 10.3390/ijerph17218112

**Published:** 2020-11-03

**Authors:** Senamile P. Ngobese, Catherine O. Egbe, Mukhethwa Londani, Olalekan A. Ayo-Yusuf

**Affiliations:** 1Alcohol, Tobacco and Other Drug Research Unit, South African Medical Research Council, Pretoria 0001, South Africa; Phindile.ngobese@mrc.ac.za (S.P.N.); Catherine.egbe@mrc.ac.za (C.O.E.); mukhethwa.londani@mrc.ac.za (M.L.); 2Department of Public Health, Sefako Makgatho Health Sciences University, Pretoria 0204, South Africa; 3Africa Centre for Tobacco Industry Monitoring and Policy Research, Sefako Makgatho Health Sciences University, Pretoria 0204, South Africa

**Keywords:** tobacco smoking, second-hand smoke, South Africa

## Abstract

Current South African tobacco control law allows for 25% designated smoking areas in some indoor public places. This study investigates non-smokers’ exposure to second-hand smoke (SHS) in workplaces, homes, cafés/restaurants, and shebeens (local bars) using data from the 2017 South African Social Attitude Survey. Factors associated with any level of exposure were explored using multiple-variable-adjusted logistic regression analysis. The sample of 3063 participants (16+ years old), comprised 51.7% females and 78.5% Black Africans. The current smoking prevalence from this study was 21.5%. About 47% of non-smokers reported exposure to SHS in at least one location. Females were significantly less likely to be exposed to SHS in all locations except at home compared to males. Adjusted logistic regression analysis showed that females, adults aged 45–54 years, 55–64 years, and 65+ years were significantly less likely to be exposed to SHS (adjusted odds ratio (AOR) = 0.63, 0.60, 0.55, and 0.24, respectively) than males and those aged 16–24 years. Those who identified as Coloureds were significantly more likely to be exposed to SHS (AOR = 1.69) than Black Africans. This study found that nearly half of non-smokers reported exposure to SHS. A 100% smoke-free policy consistent with the World Health Organisation (WHO) Framework Convention on Tobacco Control would protect more people from exposure to SHS in South Africa.

## 1. Introduction

Tobacco use is the second leading risk factor for the global burden of disease [1] and accounts for 6.3% of disability-adjusted life-years lost [2]. Second-hand smoke (SHS) is a complex toxic mixture of chemicals from the smoke that a smoker exhales or comes from the burning tip of the cigarette or other combustible tobacco products [3]. Globally, about 1.2 million non-smokers die annually as a result of diseases caused by exposure to SHS. Two-thirds of these deaths occur in developing countries, especially in Africa and Asia [4,5]. The World Health Organisation (WHO) Framework Convention on Tobacco Control (FCTC) was adopted in 2005 as a response to the globalisation of the tobacco epidemic [6,7]. Article 8 of the FCTC calls for protection from exposure to second-hand tobacco smoke and parties to the FCTC are expected to enact laws to ensure 100% smoke-free public places [8]. In 2005, South Africa became a party to the FCTC and, therefore, has a legal obligation to implement and enforce policies that protect non-smokers from involuntary exposure to SHS [9].

Currently, according to the South African Tobacco Products Act, 83 of 1993 (amended in 2008), no person may smoke any tobacco product in any indoor public place [10]. Section 2 of the act empowers the Minister of Health to make regulations regarding smoking in public places. Hence, the minister may permit smoking in the prescribed portion of a public place, subject to any prescribed condition. Therefore, in 2000, the Minister of Health published a notice in the Government Gazette relating to smoking in public places which stated that a person in control of a public place may designate a portion of the public place as a smoking area but that the designated smoking area should not exceed 25% of the total floor area of the public place [11,12]. This then allowed for certain public places to have smoking areas. The law currently states that no person is to smoke in a private dwelling that is used for commercial childcare activity (e.g., schooling and tutoring) [10]. Therefore, citizens are permitted to smoke in their homes unless they are places for childcare activities. Consequently, there is a lack of smoke-free-home campaigns to raise awareness about the harms of exposure to SHS.

According to WHO FCTC, Article 8, effective measures to provide protection from exposure to tobacco smoke require the total elimination of smoking and tobacco smoke in a particular space or environment in order to create a 100% smoke-free environment [8]. The WHO FCTC also states that there is no safe level of exposure to tobacco smoke, and notions such as a threshold value for toxicity from SHS should be rejected, as they are contradicted by scientific evidence [8]. A threshold level of toxicity can generally be understood as a value that indicates what level (threshold concentration) of exposure to a chemical could hurt people if they breathe it in or ingest it for a defined length of time (exposure duration) [13]. In other words, the current 25% designated smoking area policy in South Africa is not comprehensive enough and does not protect non-smokers from being exposed to SHS [9] or from the consequences of exposure to SHS.

Exposure to SHS is known to have adverse effects on human health. Some of these effects are cardiovascular diseases [14], strokes [15], lung cancer [16], and breast cancer [17]. In children in particular, SHS exposure causes upper and lower respiratory tract infections (e.g., bronchitis and pneumonia), asthma, otitis media (inflammation of the ear), and sudden infant death syndrome, which is the sudden unexplained death of a child of less than one year of age [18].

Evidence has shown that smoke-free ordinances inarguably and unequivocally benefit public health [19,20,21]. In addition, smoke-free laws can also help to reduce smoking prevalence in populations by encouraging smokers to quit [22]. Furthermore, these laws have the potential to raise public awareness about the dangers of tobacco smoke and can influence individuals voluntarily to adopt smoke-free home and car rules [9]. In 2018, the South African government proposed the Control of Tobacco Products and Electronic Delivery Systems Bill. Part of the bill aims to provide for 100% smokefree public places without providing for designated smoking areas [23]. Since the new tobacco control bill is still being processed, it is important to look at how effective the current laws have been in protecting non-smokers from involuntary exposure to SHS. This study theferore aims to investigate non-smokers’ exposure to SHS at selected public places and at home in South Africa.

## 2. Materials and Methods

This study draws on the data from a nationally representative sample of South African adults (16 years or older) generated by the 2017 South African Social Attitudes Survey (SASAS).

### 2.1. Survey Design and Sample

The SASAS is a household survey with a nationally representative sample. It is a cross-sectional survey that has been conducted annually by the Human Sciences Research Council (HSRC) since 2003 [24]. The survey sample was drawn from the HSRC master sample—a sampling frame that consists of 1000 Population Census enumeration areas demarcated for Census 2011. The annual surveys use a multi-stage probability sampling strategy, with census enumeration areas as the primary sampling unit. The stratification of the enumeration areas was done by the sociodemographic domains of province, geographical subtype, and the majority population group [24]. The design of the survey yielded a representative sample of adults, aged 16+ years, regardless of nationality or citizenship, in households that were geographically spread across the country’s nine provinces [24,25]. This study was approved by the HSRC ethics committee on 19 November 2016 [24].

### 2.2. Sociodemographic Characteristics

The sociodemographics evaluated in this paper included gender (female or male), self-identified race/population group (Black African (indigenous African descent), Coloured (mixed ancestry), White, or Asian/Indian), age (16–24, 25–34, 35–44, 45–54, 55–64, or 65+ years), level of education (<Grade 12, =Grade 12, or >Grade 12), marital status (married, widowed/separated/divorced, or never married), and geo-location (urban or rural).

### 2.3. Tobacco Smoking Status

Tobacco smoking status was assessed for the following products: manufactured cigarettes, roll-your-own cigarettes, hubbly–bubbly/hookah/water pipe, cigars, or pipes. Participants were independently asked if they used each of these products “currently every day”, “currently some days”, “stopped completely less than 6 months”, “stopped completely more than 6 months”, and “never smoked before”. Current tobacco smokers were participants who reported smoking any of these products every day or some days. Non-smokers were participants who reported stopping completely in less or more than 6 months and those who never smoked before.

### 2.4. Exposure to SHS

Exposure to SHS at different locations such as at home, at work, at a café/restaurant, or shebeen (local bar) was assessed separately by the following question: “In the past 30 days, about how many days would you say you were in a place where someone smoked close to you (no complete physical barrier, i.e., smoke got to you)”? The response categories were “Never”, “1–6 days”, “7–10 days”, “11–15 days”, “16–20 days”, “more than 20 days”, and “Refused to answer”. All responses other than “Never” and “Refused to answer” were categorised as being exposed to SHS in the respective locations assessed. Any location means being exposed in at least one of the assessed locations. Participants exposed in only one, two, three, or all four locations being assessed were graded 1, 2, 3, and 4, respectively. Frequency of exposure was graded as “Never” = 0, “1–6 days” = 1, “7–10 days” = 2, “11–15 days” = 3, “16–20 days” = 4, and “more than 20 days” = 5. Level of exposure was determined by computing grades for location of exposure and frequency of exposure resulting in a scale ranging from 0 to 8. The level of exposure scale was then categorised as; 0 = 0 (no level of exposure), 1–2 = 1 (low), 3–4 = 2 (moderate), 5–6 = 3 (high), and 7–8 = 4 (very high) level of exposure.

### 2.5. Data Analysis

All data were weighted to account for the complex survey design and to yield nationally representative estimates. Percentages, chi square, and frequencies were calculated to describe the general sample. The proportion of non-smokers exposed to SHS, and the proportion of non-smokers exposed to SHS per location (home, work, café/restaurant, and shebeen) and level of exposure was calculated by age, gender, race, highest level of education, marital status, and geo-location. Factors associated with any level of exposure was explored using multiple-variable-adjusted logistic regression analysis. Data were analysed using STATA version 15, StataCorp, College Station, TX, USA. All analyses involving exposure to SHS were conducted for non-smokers only.

## 3. Results

The sample was made up of 3063 participants aged 16+ years and above. About 52% (*n* = 1864) were female; 79% (*n* = 1872) were of Black African descent, and approximately 26% (*n* = 637) of the sample were 25–34 years old. The majority, 54% (*n* = 1660), of the sample had an education level less than Grade 12; about 51% (*n* = 1189) were never married; most participants (69%, *n* = 2324) were from the urban area; and about 21% (*n* = 651) of the sample were current smokers (smoked combustible tobacco products) (see Table 1 for a summary of the demographics).

### 3.1. Non-Smokers’ Exposure to Tobacco Smoke at Home, Work, and Hospitality Venues

Among non-smokers, about 47% reported exposure to SHS from at least one location. Regarding specific sources of exposure, 26% reported being exposed to SHS at home, 22% were exposed in a café/restaurant, 19% at a shebeen, and 13% at work. In terms of gender, males were significantly more exposed to SHS than females: at work (19.6% vs. 8.8%), in cafés/restaurants (29.1% vs. 17.3%), and at shebeens (26.5% vs. 13.5%). However, males were less exposed to SHS at home than females (23.8% vs. 27.2%), but this association was not significant. Exposure to SHS by race displayed some significant differences: those who identified as Coloured were significantly more exposed than were other race groups at home (39.3%), while Black Africans were significantly more exposed at shebeens (20.4%). In all other places, race was not significantly associated with exposure to SHS.

Exposure by age differed: individuals in the age group of 35 to 44 years were significantly more exposed to SHS than were other age groups at work (18.1%), while persons in the age category 16 to 24 years were significantly more exposed than other age categories at cafés/restaurants (30.0%) and at shebeens (24.4%). Participants between the ages of 16 and 24 years were more exposed to SHS at home, but the association was not significant. Results on the exposure to SHS by level of education revealed that participants who attained Grade 12 education were significantly more exposed to SHS at home (29.6%) and at shebeens (22.9%). Individuals with more than a Grade 12 level of education were significantly more exposed at work (23.8%) and at cafés/restaurants (28.7%). Exposure by marital status revealed differences too: participants who have never been married were significantly more exposed to SHS at home (29.9%), at cafés/restaurants (24.1%) and at shebeens (21.9%), while those who were married were significantly more exposed at work (16.5%) (see Table 2).

### 3.2. Level of Exposure to SHS by Demographic Characteristics

About 27% of the study sample reported low level of exposure, while 14.7% reported moderate exposure. Only 6% of the sample reported a high to very high level of exposure to SHS. Approximately 8.5% of male participants experienced high to very high exposure to SHS. In terms of race, 6.4% of Black Africans experienced high to very high levels of exposure while 8.1% of individuals who were between the ages 16 and 24 years experienced a high to very high level of exposure. Among those with Grade 12 as their highest level of education, 8.2% reported that they had experienced high to very high levels of exposure to SHS compared to 4.6% of those with less than Grade 12. About 7.4% of individuals who were never married experienced high to very high levels of exposure to SHS. Geo-location was not significantly associated with the level of exposure of participants to SHS (see Table 3).

### 3.3. Any Level of Exposure to SHS

When investigating any level of exposure to SHS (not exposed vs. exposed (low to very high exposure)) in a multiple variable regression model, it was found that females (Adjusted Odds Ratio (AOR) = 0.63; 95% Confidence Interval (CI): 0.47–0.86) were significantly less likely to be exposed to SHS than were males. Compared to Black Africans, the odds of any level of exposure to SHS of those who self-identified as Coloured were significantly higher (AOR = 1.69; 95% CI: 1.11–2.57). Compared to those that were 16 to 24 years of age, the odds of being exposed to SHS were significantly lower for adults aged 45 to 54 years (AOR = 0.60; 95% CI: 0.36–1.00), 55 to 64 years (AOR = 0.55; 95% CI: 0.33–0.91), and 65+ years and above (AOR = 0.24; 95% CI: 0.14–0.41). All other demographic characteristics were not significantly associated with level of exposure to SHS (see Table 4).

## 4. Discussion

The objective of this study was to investigate non-smokers’ exposure to SHS at selected indoor public venues and at home. We also explored the sociodemographic factors that are associated with SHS exposure at these venues. Our results show that non-smokers’ exposure to tobacco smoke is high and is a cause for concern in South Africa. However, the about 47% of non-smokers exposed to SHS obtained in our study indicates a decrease compared to the 55.9% reported in a similar study done in 2010 [9]. Such a decrease may probably be ascribed to the tobacco control efforts that have led to people becoming more aware of the dangers of tobacco use and, therefore, to more people adopting a healthier lifestyle by quitting smoking or smoking less in public places [9]. Worryingly, most participants were found to be exposed to SHS at home, and the majority of the people exposed there were the youngest in terms of the age groups. These findings are of concern because studies have shown that children or young people exposed to SHS at home are more likely to smoke in the future, due to the normalisation of smoking by their significant others [26]. Our findings indicate that participants who are 16 to 24 years of age experience very high exposure to SHS. These results emphasise the urgent need for South Africa to raise public awareness of the dangers of SHS and implement laws that would encourage more adults who smoke to quit, which would in turn lead to reduced smoking at home.

In a study assessing the risk of burden of disease and injury attributable to 67 risk factors, it was found that for people as young as 15+ years, the leading risk factor for death and disease worldwide was alcohol use, followed by tobacco smoking, including exposure to SHS [2]. Findings from our study reveal that more young people are exposed to SHS at hospitality venues (café/restaurant and shebeens). When more young people are exposed to SHS, this can lead to a sicklier population in the future, putting the economy of the country at risk. Diseases caused by smoking are a liability to the country’s economy—in 2012, these diseases accounted for 5.7% of global health expenditure [27]. Therefore, the implementation of a 100% smoke-free policy will not only lead to a lower percentage of people exposed to SHS, which in turn would result in improved overall health, but it will also result in a decrease in smoking prevalence (by discouraging initiation and encouraging quitting) and, consequently, it will have a positive economic impact.

A retrospective study on the worldwide burden of disease from exposure to SHS found that in places such as Africa, some parts of the Americas, the eastern Mediterranean, and southeast Asia, women are at least 50% more likely to be exposed to SHS than men are [1]. However, our results showed that females were significantly less likely to be exposed to SHS in all locations except at home compared to males. The South African data are however in line with those of studies done in China [28] and in Ethiopia [29], which report similar findings of women having higher exposure to SHS and at home. Unfortunately, the instrument used in this study was not developed to explain why women are mostly exposed at home. Possible explanations for this finding could be that other family members are probably smokers and women mostly do not have the authority to make rules such as “no smoking at home” in the family. Moreover, our results show that males were more exposed to SHS at work, in cafés/restaurants, and at shebeens than females were. The higher percentage of non-smoking men exposed to SHS at work may be due to their greater interaction with other male workers, many of whom probably smoke [30], and the same reason may apply to why in hospitality places males are more exposed than females are. Moreover, our results show that more males experience a very high level of exposure overall. The prevalence of smoking in South Africa over the years has been higher among males than among females [31,32]. Therefore, due to a greater interaction with smokers, it is logical that males would experience higher levels of exposure to SHS.

We found a significant relationship between self-identified race and exposure to SHS in this study. Participants of African descent were found to be more exposed at shebeens, which are mostly located in traditionally Black townships [33]. Participants who identified as White were mostly exposed in cafés/restaurants, although the finding is not statistically significant. As opposed to shebeens, cafés/restaurants are mostly situated in cities and suburban places. Implementing a 100% smoke-free policy would not only benefit those who visit public places but would also protect those employed at such places as well. Our findings show that 13% of the participants were exposed at work, which underscored the plea for awareness regarding the Government Gazette notice related to smoking tobacco products in public places, which states that employees may object to tobacco smoke in the workplace without retaliation of any kind [11]. For this reason, it is imperative for both employers and employees to be included in interventions for tobacco control.

Individuals who self-identified as Coloured experienced the highest level of exposure to SHS. Our findings are similar to those of a previous study done in South Africa, which found that more Coloureds reported more exposure to SHS than members of other race groups [32]. Our study did not explore the reason for the high exposure among Coloureds, but the prevalence of smoking in South Africa is highest among the Coloured population than among other race groups [31]. More research is needed to explore this finding in order to understand tobacco use behaviour in this subpopulation.

Limitations of this study include the fact that it is based on a cross-sectional survey that relies on self-reports. There is a possibility that socially desirable responses were given and that there might have been recall bias.

## 5. Conclusions

Current South African tobacco control legislation allows for designated smoking areas in some indoor public places, with the result that nearly half of the non-smoking sample reported being exposed to SHS in public places and at home. In particular, more young people are exposed to SHS. Inevitably, the absence of policies to protect people, particularly the youth, from being exposed to SHS leads to an unhealthy population. An unhealthy population in turn burdens the country’s economy in healthcare costs and loss of productivity. This makes a 100% smoke-free policy the desired standard for smoke-free policies to ensure the protection of non-smokers from SHS. Furthermore, strong enforcement of such smoke-free laws could help to increase compliance and to denormalise smoking, thus reducing smoking prevalence in the population.

## Figures and Tables

**Table 1 ijerph-17-08112-t001:** Demographic characteristics and smoking status of participants.

Demographic and Smoking Characteristics	*n* (%) *	95% CI
Gender		
Female	1864 (51.8)	48.5–55.0
Male	1199 (48.2)	44.9–51.5
Race		
Black African	1872 (78.5)	74.6–81.9
Coloured	495 (9.0)	7.1–11.4
White	348 (9.6)	7.4–12.5
Asian/Indian	348 (2.8)	2.1–3.9
Age		
16–24	498 (24.2)	21.8–26.7
25–34	637 (26.5)	24.2–28.9
35–44	613 (19.1)	17.4–20.8
45–54	437 (13.4)	11.7–15.4
55–64	413 (9.3)	8.1–10.7
65+	465 (7.6)	6.4–9.0
Highest Level of Education		
<Grade 12	1660 (54.3)	50.9–57.6
Grade 12	895 (33.2)	30.5–36.1
>Grade 12	441 (12.6)	10.5–14.9
Marital Status		
Married	1254 (38.6)	35.9–41.3
Widowed/divorced/separated	507 (10.3)	8.9–11.8
Never married	1189 (51.2)	48.3–54.1
Geo-location		
Urban	2324 (69.2)	63.4–74.5
Rural	739 (30.8)	25.6–36.6
Smoking Status		
Current smoker	651 (21.1)	18.9–23.5
Non-smoker	2367 (78.9)	76.5–81.1

CI – Confidence Interval; * *n* = 3063.

**Table 2 ijerph-17-08112-t002:** Non-smokers’ exposure to second-hand smoke (SHS) at home, work and hospitality venues.

Demographic Characteristics	At Home		At Work		At Café/Restaurant		At Shebeen		Any Location	
	Exposed %	*p*-Value	Exposed %	*p*-Value	Exposed %	*p*-Value	Exposed %	*p*-Value	Exposed %	*p*-Value
Total *n* (%)	1000 (25.9)		482 (13.1)		726 (22.0)		658 (18.8)		1573 (46.6)	
Gender		0.332		<0.001 *		<0.001 *		0.001 *		0.006 *
Female	27.2 (23.6–31.3)		8.8 (6.8–11.2)		17.3 (14.6–20.3)		13.5 (11.0–16.4)		42.5 (38.5–46.6)	
Male	23.8 (18.4–30.3)		19.6 (15.3–24.8)		29.1 (24.2–34.5)		26.5 (21.5–32.2)		52.8 (46.6–58.8)	
Race		0.006 *		0.303		0.081		0.001 *		0.223
Black African	25.7 (21.9–30.0)		12.7 (10.3–15.7)		20.9 (18.0–24.1)		20.4 (17.4–23.7)		46.1 (42.1–50.2)	
Coloured	39.3 (30.0–49.4)		12.3 (8.4–17.7)		27.3 (20.9–34.8)		12.1 (8.0–17.8)		55.6 (46.8–64.0)	
White	17.4 (11.2–26.0)		17.8 (12.2–25.3)		28.6 (21.0–37.6)		12.6 (7.6–20.2)		43.9 (34.8–53.5)	
Asian/Indian	22.8 (13.1–36.7)		13.2 (7.2–22.9)		24.0 (15.3–35.50		3.9 (1.8–8.2)		47.9 (37.5–58.6)	
Age		0.085		0.007 *		0.001 *		0.001 *		<0.001 *
16–24	30.4 (23.7–38.1)		10.6 (6.8–16.2)		30.0 (23.9–36.9)		24.4 (19.0–30.7)		54.6 (47.2–61.7)	
25–34	28.1 (21.7–35.6)		16.1 (11.7–21.8)		21.7 (17.1–27.0)		23.3 (17.7–29.9)		50.2 (43.3–57.1)	
35–44	24.5 (18.6 -31.6)		18.1 (12.6–25.2)		21.5 (16.4–27.7)		20.0 (15.1–26.0)		47.9 (41.1–54.7)	
45–54	24.5 (17.8–32.6)		14.0 (9.3–20.5)		21.7 (15.2–30.0)		13.5 (8.3–21.2)		41.2 (33.6–49.2)	
55–64	21.7 (15.6–29.4)		8.7 (5.2–14.4)		13.7 (9.3–19.9)		8.4 (4.1–16.4)		40.0 (32.6–47.8)	
65+	14.9 (10.6–20.5)		3.1 (1.1–8.5)		11.2 (7.4–16.5)		5.0 (2.6–9.5)		24.1 (18.9–30.0)	
Highest Level of Education		0.009 *		<0.001 *		0.037 *		0.021 *		0.041 *
<Grade 12	26.5 (22.8–30.6)		8.6 (6.5–11.4)		19.1 (15.9–22.8)		15.3 (12.4–18.8)		43.6 (39.4–47.9)	
Grade 12	29.6 (23.3–36.7)		15.4 (11.5–20.2)		23.0 (18.8–27.8)		22.9 (19.1–27.3)		52.1 (46.1–58.0)	
>Grade 12	14.8 (9.8–21.6)		23.8 (16.4–33.3)		28.7 (21.5–37.1)		21.8 (14.7–31.2)		44.3 (36.2–52.8)	
Marital Status		0.016 *		0.013 *		0.097		0.026 *		0.036 *
Married	23.7 (19.3–28.9)		16.5 (13.4–20.2)		20.8 (16.7–25.5)		16.9 (12.6 -22.2)		45.7 (40.7–50.8)	
Widowed/Divorced/Separated	18.3 (13.3–24.6)		7.2 (4.0–12.8)		15.2 (10.4–21.6)		10.3 (5.8–17.6)		37.5 (30.6–44.9)	
Never Married	29.9 (25.0–35.3)		11.5 (8.5–15.5)		24.1 (20.2–28.4)		21.9 (18.3–25.9)		49.9 (44.9–55.0)	
Geo-location		0.429		0.231		0.128		0.807		0.396
Urban	26.8 (22.5–31.5)		14.1 (11.5–17.2)		20.5 (17.8–23.5)		18.6 (15.5–22.0)		47.7 (43.4–51.9)	
Rural	23.9 (19.1–29.6)		11.0 (7.7–15.6)		25.3 (20.0–31.6)		19.3 (14.7–24.9)		44.4 (38.4–50.6)	

** p* significant at ≤ 0.05.

**Table 3 ijerph-17-08112-t003:** Non-smokers’ level of exposure to SHS by demographics.

Demographic Characteristics	No Exposure % (95% CI)	Low Exposure % (95% CI)	Moderate Exposure % (95% CI)	High Exposure % (95% CI)	Very High Exposure % (95% CI)	*p*-Value
Total *n* (%)	1398 (52.7)	767 (26.6)	544 (14.7)	207 (4.1)	55 (1.9)	
Gender						0.001 *
Female	57.0 (52.8–61.1)	27.6 (24.1–31.4)	11.1 (9.1–13.4)	3.0 (2.0–4.6)	1.4 (0.6–3.1)	
Male	46.2 (40.1–52.4)	25.1 (20.4–30.6)	20.2 (16.2–25.0)	5.7 (3.7–8.8)	2.7 (1.2–6.1)	
Race						0.017 *
Black African	53.3 (49.2–57.4)	26.4 (23.0–30.0)	14.0 (11.6–16.8)	4.1 (2.7–6.0)	2.3 (1.3–4.1)	
Coloured	41.4 (33.2–50.2)	34.6 (26.4–43.8)	20.0 (14.2–27.3)	3.7 (2.0–6.9)	0.3 (0.1–2.4)	
White	56.0 (46.4–65.1)	19.0 (13.0–26.9)	20.2 (13.5–29.1)	4.9 (2.1–10.8)	0.1 (0.0–0.4)	
Asian/Indian	51.4 (40.7–62.0)	38.3 (27.6–50.2)	6.8 (3.4–13.0)	3.5 (1.4–8.9)	0	
Age						0.001 *
16–24	45.1 (37.9–52.5)	31.6 (24.5–39.7)	15.3 (10.8–21.2)	3.8 (2.1–6.8)	4.3 (1.8–9.7)	
25–34	49.2 (42.3–56.2)	25.8 (19.9–32.7)	18.7 (14.5–23.8)	5.5 (2.8–10.5)	0.8 (0.3–2.3)	
35–44	50.5 (43.7–57.4)	26.7 (21.1–33.1)	16.9 (12.4–22.6)	3.7 (1.7–7.6)	2.2 (0.7–6.7)	
45–54	57.9 (49.7–65.7)	21.0 (15.4–28.0)	13.5 (9.3–19.4)	6.1 (2.3–15.0)	1.5 (0.4–5.6)	
55–64	59.7 (51.8–67.1)	31.1 (24.2–38.9)	7.3 (4.0–12.8)	2.0 (0.9–4.3)	0	
65+	75.6 (69.6–80.8)	17.5 (13.3–22.7)	5.4 (2.9–9.9)	1.0 (0.2–3.8)	0.5 (0.1–3.0)	
Highest Level of Education						0.004 *
<Grade 12	55.7 (51.4–60.0)	27.8 (23.9–32.0)	11.9 (9.6–14.8)	3.1 (1.9–5.1)	1.5 (0.7–3.2)	
Garde 12	47.6 (41.6–53.6)	28.8 (23.5–34.9)	15.5 (11.9–19.9)	5.0 (3.0–8.1)	3.2 (1.4–6.8)	
>Grade 12	55.6 (47.1–63.7)	16.2 (11.7–21.9)	21.8 (15.2–30.3)	6.0 (2.7–13.0)	0.5 (0.1–3.2)	
Marital Status						0.008 *
Married	53.7 (48.6–58.8)	24.5 (21.1–28.3)	16.5 (12.8–21.0)	4.5 (2.4–8.2)	0.8 (0.3–1.9)	
Widowed/Divorced/Separated	61.9 (54.2–68.9)	27.4 (21.1–34.8)	9.6 (5.9–15.2)	1.0 (0.3–3.3)	0.2 (0.0–0.9)	
Never Married	49.6 (44.5–54.7)	28.7 (24.1–33.7)	14.4 (11.4–17.9)	4.2 (2.7–6.5)	3.2 (1.6–6.1)	
Geo-location						0.760
Urban	51.8 (47.5–56.1)	27.6 (24.0–31.6)	15.1 (12.5–18.0)	3.9 (2.5–6.0)	1.6 (0.8–3.5)	
Rural	54.6 (48.4–60.7)	24.4 (19.8–29.7)	14.0 (10.3–18.8)	4.5 (2.6–7.6)	2.5 (1.0–5.8)	

CI- Confidence Interval; ** p* significant at ≤ 0.05.

**Table 4 ijerph-17-08112-t004:** Multiple-variable-adjusted logistic regression for any level of exposure to SHS.

Co-Variates	AOR	95% CI	*p*-Value
Gender			
Female	0.63	(0.47–0.86)	0.003 *
Male	Ref.		
Race			
Black African	Ref.		
Coloured	1.69	(1.11–2.57)	0.014 *
White	0.90	(0.55–1.48)	0.677
Asian/Indian	1.14	(0.69–1.89)	0.615
Age			
16–24	Ref.		
25–34	0.92	(0.62–1.36)	0.657
35–44	0.82	(0.53–1.25)	0.347
45–54	0.60	(0.36–1.00)	0.048 *
55–64	0.55	(0.33–0.91)	0.021 *
65+	0.24	(0.14–0.41)	<0.001*
Education			
<12 Grade 12	Ref.		
Grade 12	1.07	(0.78–1.47)	0.672
>Grade 12	0.81	(0.53–1.23)	0.315
Marital Status			
Married	Ref.		
Widowed/Divorced/Separated	1.19	(0.79–1.81)	0.406
Never Married	0.94	(0.69–1.27)	0.670
Geo-location			
Urban	Ref.		
Rural	0.92	(0.67–1.26)	0.584

AOR – Adjusted Odds Ratio; CI – Confidence Interval; ** p* significant at ≤ 0.05.

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
