# Peer review of "Non-Smoker’s Exposure to Second-Hand Smoke in South Africa during 2017"

_ijerph, 2020, doi:10.3390/ijerph17218112_

Round 1
Reviewer 1 Report
Dear Editors and the Authors,
Thank you for the invitation to review the manuscript entitled “Non-smoker’s exposure to second-hand smoke in South Africa during 2017” prepared by Senamile Ngobese and colleagues.
The manuscript reports from a cross-sectional 2017 study among a representative sample from South Africa. The authors analyse the level of exposure to second hand smoke (SHS) and their correlates among non-smokers aged 16+. The manuscript is written very well and clearly. It is an interesting and important study to report.
However, I have some suggestions for how this analysis and manuscript presentation could be improved and would be very interested to learn the Authors’ perspective on these:
Abstract:
- I think ‘and’ should be between restaurant and shebeens: “cafés and restaurants, shebeens”. Also, after reading the rest of the article it seems that restaurants are listed by a mistake?
- The sample was quite even for male and female gender (so majority was not female?)
- From the abstract it is not clear how the smoking prevalence was calculated – is it from the same study or national statistics? If it’s the latter, then this should be at the start of the abstract, not with the study results. If it’s a study result then it should be mentioned in the methods, e.g. ‘Smoking prevalence, and factors associated with SHS exposure and SHS levels among non-smokers were explored…’. Then the abstract could also read (but not that needed if it's a representative sample): ‘in the present study smoking prevalence was 21.5%, but …’ or similar.
- Lines 42-43 report about SA tobacco control law – this should be at the start of the abstract, as setting the scene/background.
Methods
- I suggest to separate the assessment of smoking status from that of exposure to SHS (at the moment they are under the same sub-heading 2.3, line 114).
- How were non-smokers classified?
- Assessment of SHS exposure – it is not clear how/when/where the location (e.g. workplace) was mentioned with this question: “In the past 30 days, about how many days would you say you were in a place where someone smoked close to you (no complete physical barrier, i.e. smoke got to you)? The current wording suggests that the frequency of SES exposure was assessed in general and not for specific locations.
- Where smokers also asked the same question on SHS exposure as non-smokers?
- It would be interesting to compare data on SHS exposure among smokers and non-smokers (if this data was collected).
- There are many analyses but no corrections for familywise error (e.g. I am not sure if p-values of 0.021 or 0.026 should still be considered significant?).
- Data analysis – it is quite clear how the authors arrived at 0-4 classification of exposure, but this doesn’t account for frequency of exposure? Current labels is quite misleading. So someone could be exposure for only 1-6 days in each location and get 4 points, and someone else be exposure more than 20 days in 2 locations and get only 2 points. This doesn’t sound right. Perhaps the frequency of exposure (i.e. days) should be converted to continuous, e.g. never=0 days, 1-6 = 3 days, 7 to 10 = 8.5 days etc, and then multiply the continuous age by the number of locations to get a better indication of the level of exposure.
- If the authors do not want to incorporate the frequency of exposure (which is a considerable limitation or at least a missed opportunity, I think), then I think this needs to be stated clearly in the methods and limitations, and a good rationale provided for this decision. Also, I would not use the label ‘level of exposure’ but something like ‘number of locations with SHS exposure.
- Data analysis mentions how key variables were created (e.g. exposure level with 4 categories). However, the information or variables, especially dependent variables are really crucial and it would be very helpful if it was listed under the variables (under the separate sub-heading for exposure to SHS) and not in the analysis.
- Another dependent variable (in Table 3) is ‘any exposure to SHS’. This should be again listed and explained together with exposure level under the variable list.
Results
- From the tables it is not clear if the analysis is adjusted or unadjusted (this could be in the title of the table), and if it adjusted, what is it adjusted for? This could be listed in the tables’ footnotes.
- Table 2 is missing category ‘restaurants’ that are listed in the abstract. Is there a mistake in the abstract?
- Table 4 needs a footnote to clarify/remind the readers what does no exposure (0 locations with SHS exposure), low exposure (1 location with SHS exposure….) mean.
- Table 5- it is unclear why this analysis was conducted? It seems to tell the same results as Table 4? The decision which one to run and report should be made before the analysis and it should be explained which one was selected to run and to report.
Discussion
It is written very well, but some things may change if some of the changes to the analysis/results are implemented.
19. Limitations could be expanded - depending on which suggestions will be implemented.
Conclusions
20. I feel the last paragraph is good but could be improved – rewritten in a way to make a stronger focused case.
Author Response
IJERPH First Reviewer
Dear Editors and the Authors,
Thank you for the invitation to review the manuscript entitled “Non-smoker’s exposure to second-hand smoke in South Africa during 2017” prepared by Senamile Ngobese and colleagues.
The manuscript reports from a cross-sectional 2017 study among a representative sample from South Africa. The authors analyse the level of exposure to second hand smoke (SHS) and their correlates among non-smokers aged 16+. The manuscript is written very well and clearly. It is an interesting and important study to report.
Response: Thank you
However, I have some suggestions for how this analysis and manuscript presentation could be improved and would be very interested to learn the Authors’ perspective on these:
Abstract:
Reviewer 1:
- I think ‘and’ should be between restaurant and shebeens: “cafés and restaurants, shebeens”. Also, after reading the rest of the article it seems that restaurants are listed by a mistake?
Response:
We have now added “and” between restaurants and sheebens’. Café and restaurant are grouped together as one location as these are just different variants of an eating place, so we have now added ‘/’ in between café and restaurant throughout the manuscript.
Reviewer 1:
- The sample was quite even for male and female gender (so majority was not female?)
Response:
Thank you. We have now removed the word “mainly”.
Reviewer 1:
- From the abstract it is not clear how the smoking prevalence was calculated – is it from the same study or national statistics? If it’s the latter, then this should be at the start of the abstract, not with the study results. If it’s a study result then it should be mentioned in the methods, e.g. ‘Smoking prevalence, and factors associated with SHS exposure and SHS levels among non-smokers were explored…’. Then the abstract could also read (but not that needed if it's a representative sample): ‘in the present study smoking prevalence was 21.5%, but …’ or similar.
Response:
We have now indicated that the current smoking prevalence is from this study, it now reads” Current smoking prevalence from this study was 21.5%...”
Reviewer 1:
- Lines 42-43 report about SA tobacco control law – this should be at the start of the abstract, as setting the scene/background.
Response:
Thank you. We have now moved the report about the South African tobacco law to the start of the abstract as suggested.
Reviewer 1:
Methods
- I suggest to separate the assessment of smoking status from that of exposure to SHS (at the moment they are under the same sub-heading 2.3, line 114).
Responses:
We have now separated the tobacco smoking status from exposure to SHS. Please see page 4.
Reviewer 1:
- How were non-smokers classified?
Response:
Thank you. We have included the definition of non-smokers, it reads “Non-smokers were participants who reported stopping completely in less or more than 6 months and never smoked before.”
Reviewer 1:
- Assessment of SHS exposure – it is not clear how/when/where the location (e.g. workplace) was mentioned with this question: “In the past 30 days, about how many days would you say you were in a place where someone smoked close to you (no complete physical barrier, i.e. smoke got to you)? The current wording suggests that the frequency of SES exposure was assessed in general and not for specific locations.
Response:
Thank you. We have now indicated that this question was asked independently for all four locations under investigation.
Reviewer 1:
- Where smokers also asked the same question on SHS exposure as non-smokers?
Response:
In the larger study, every participant was asked this question. However, in this paper we correctly focused only on non-smokers’ responses.
Reviewer 1:
- It would be interesting to compare data on SHS exposure among smokers and non-smokers (if this data was collected).
Response:
Thank you. However, this is out of scope of this paper.
Reviewer 1:
- There are many analyses but no corrections for familywise error (e.g. I am not sure if p-values of 0.021 or 0.026 should still be considered significant?).
Response:
We do not believe corrections for familywise error is applicable to our analyses. All analyses were conducted to measure similar outcomes variables. Please see our methods section for how the outcome variables were created.
Reviewer 1:
- Data analysis – it is quite clear how the authors arrived at 0-4 classification of exposure, but this doesn’t account for frequency of exposure? Current labels is quite misleading. So someone could be exposure for only 1-6 days in each location and get 4 points, and someone else be exposure more than 20 days in 2 locations and get only 2 points. This doesn’t sound right. Perhaps the frequency of exposure (i.e. days) should be converted to continuous, e.g. never=0 days, 1-6 = 3 days, 7 to 10 = 8.5 days etc, and then multiply the continuous age by the number of locations to get a better indication of the level of exposure.
Response:
We have now re-assessed level of exposure to include frequency of exposure and number of locations exposed at. Our initial variable referred to as “level of exposure” has been changed to “locations of exposure” Tables 3 to 5 reflect results based on the new “level of exposure” variable as DV. However, the new analyses conducted did not present a significant change in the old results. See tables 3 to 5
Reviewer 1:
- If the authors do not want to incorporate the frequency of exposure (which is a considerable limitation or at least a missed opportunity, I think), then I think this needs to be stated clearly in the methods and limitations, and a good rationale provided for this decision. Also, I would not use the label ‘level of exposure’ but something like ‘number of locations with SHS exposure.
Response:
We have changed this to locations where non-smokers were exposed to SHS and re-assessed level of exposure using frequency and location variables. Thank you.
Reviewer 1:
- Data analysis mentions how key variables were created (e.g. exposure level with 4 categories). However, the information or variables, especially dependent variables are really crucial and it would be very helpful if it was listed under the variables (under the separate sub-heading for exposure to SHS) and not in the analysis.
Response:
Dependent variables are now presented separately (please see page 4).
Reviewer 1:
- Another dependent variable (in Table 3) is ‘any exposure to SHS’. This should be again listed and explained together with exposure level under the variable list.
Response:
Thank you. We have added explanations about these variables (Any level of exposure and exposure at any location) in the method section.
Reviewer 1:
Results
- From the tables it is not clear if the analysis is adjusted or unadjusted (this could be in the title of the table), and if it adjusted, what is it adjusted for? This could be listed in the tables’ footnotes.
Response:
This is now indicated in the titles of the tables. Please see Tables 3 and 5.
Reviewer 1:
- Table 2 is missing category ‘restaurants’ that are listed in the abstract. Is there a mistake in the abstract?
Response:
Thank you. We have now added restaurants to the table, cafés and restaurants were grouped as one category in this study.
Reviewer 1:
- Table 4 needs a footnote to clarify/remind the readers what does no exposure (0 locations with SHS exposure), low exposure (1 location with SHS exposure….) mean.
Response:
Table 4 now presents results for the new level of exposure variable which combines locations and frequency of exposure so this suggested edit has been taken care of.
Reviewer 1:
- Table 5- it is unclear why this analysis was conducted? It seems to tell the same results as Table 4? The decision which one to run and report should be made before the analysis and it should be explained which one was selected to run and to report.
Response:
Thank you. Table 5 has now been deleted. Also, we moved Table 4 to become Table 3 and Table 3 is now Table 4 to allow for easy flow of the results section.
Reviewer 1:
Discussion
It is written very well, but some things may change if some of the changes to the analysis/results are implemented.
Response:
Thank you. The changes in the analysis did not change the results significantly hence, the discussion did not change.
Reviewer 1:
- Limitations could be expanded - depending on which suggestions will be implemented.
Response:
No new limitation statement was included since suggestions which would have changed the limitations were all implemented.
Reviewer 1:
Conclusions
- I feel the last paragraph is good but could be improved – rewritten in a way to make a stronger focused case.
Response:
Thank you. We have reviewed the conclusion.
Reviewer 2 Report
I have read this manuscript with great interest. SHS is an important issue that affects the health of many non-smokers globally. The importance of implementing smoke-free places policies across the world is apparent.
Overall, I found this to be a well thought-out and clearly presented manuscript.
I only have a few minor points:
Line 62 – does it mean public places including outdoors, such as parks, or indoor only?
The authors provide background on smoking rules in public places. Could you please elaborate on smoke-free homes policies and campaigns (or lack thereof)?
Discussion – 3rd paragraph – states females had higher odds of being exposed to SHS than males – this is at odds with the rest of the paper, which suggests that women are less exposed (OR=0.64), except at home, but that was non-significant. (was this supposed to say “at home”?)
Author Response
IJERPH Second Reviewer
Reviewer 2:
I have read this manuscript with great interest. SHS is an important issue that affects the health of many non-smokers globally. The importance of implementing smoke-free places policies across the world is apparent.
Overall, I found this to be a well thought-out and clearly presented manuscript.
I only have a few minor points:
Response: Thank you
Reviewer 2:
Line 62 – does it mean public places including outdoors, such as parks, or indoor only?
Response:
Thank you. We have indicated that the public places referred to are indoor public places only.
Reviewer 2:
The authors provide background on smoking rules in public places. Could you please elaborate on smoke-free homes policies and campaigns (or lack thereof)?
Response:
Thank you. We have added “The law currently states no person is to smoke in a private dwelling that is used for commercial childcare activity (e.g. schooling and tutoring) [10]. Therefore, citizens are permitted to smoke in their homes unless they are places of childcare activities. Consequently, there is a lack of smoke-free home campaigns to raise awareness about the harms of exposure to SHS.
Reviewer 2:
Discussion – 3rd paragraph – states females had higher odds of being exposed to SHS than males – this is at odds with the rest of the paper, which suggests that women are less exposed (OR=0.64), except at home, but that was non-significant. (was this supposed to say “at home”?)
Response:
We have revised the sentence to read “However, our results showed that females were significantly less likely to be exposed to SHS in all locations except at home compared to males.”